# Impact of Inherited Thrombophilia on Pregnancy Complications and the Role of Low-Molecular-Weight Heparin Therapy: A Case–Control Study

**DOI:** 10.3390/medicina61071131

**Published:** 2025-06-24

**Authors:** Dragana Maglic, Vesna Mandic-Markovic, Zeljko Mikovic, Rastko Maglic, Radomir Anicic, Milica Mandic

**Affiliations:** 1Faculty of Medicine, University of Belgrade, Dr Subotica 8, 11000 Belgrade, Serbia; vesna.mandic-markovic@med.bg.ac.rs (V.M.-M.); zeljko.mikovic@med.bg.ac.rs (Z.M.); rastko.maglic@med.bg.ac.rs (R.M.); radomir.anicic@med.bg.ac.rs (R.A.); 2Department for High-Risk Pregnancies, University Clinic for Gynecology and Obstetrics “Narodni Front”, Kraljice Natalije 62, 11000 Belgrade, Serbia; mandic.milica@gakfront.org

**Keywords:** inherited thrombophilia, pregnancy complications, low-molecular-weight heparin, factor V Leiden, fetal growth restriction risk stratification

## Abstract

*Background and Objectives*: Inherited thrombophilia (IT) increases the risk of adverse pregnancy outcomes, but the benefit of low-molecular-weight heparin (LMWH) prophylaxis remains debated. This study aimed evaluate the effect of LMWH by analyzing outcomes in women with IT who received LMWH versus those who did not and also compare pregnancy complication rates before and after inherited thrombophilia diagnosis. *Materials and Methods*: We conducted a retrospective case–control study including 276 pregnant women with inherited thrombophilia and prior pregnancy complications and 276 healthy pregnant controls on delivery. The main outcome was the incidence of complications: preterm rupture of membranes, oligohydramnios, fetal growth restriction, preterm delivery, stillbirth, HELLP syndrome, gestational hypertension, deep vein thrombosis, and recurrent pregnancy loss. The effect of LMWH was assessed by comparing complication rates among inherited thrombophilia patients who received therapy versus those who did not. *Results*: Women with IT were older, had higher BMI, delivered earlier, and had neonates with lower birth weight compared to controls. In current pregnancies, LMWH was associated with reduced rates of preterm delivery, fetal growth restriction, gestational hypertension, and recurrent pregnancy loss, especially in factor V Leiden carriers. LMWH had little effect on low-risk genotypes and was not independently associated with outcome reduction. *Conclusions*: LMWH prophylaxis should be reserved for high-risk women with IT. Routine use in all IT pregnancies is not justified and may cause unnecessary risks and costs. Early screening, risk stratification, and individualized care are essential to optimize outcomes.

## 1. Background and Objectives

Inherited thrombophilia (IT) refers to genetic conditions that increase the risk of deep vein thrombosis (DVT) and pregnancy complications (PCs), such as recurrent pregnancy loss (RPL), gestational hypertension (GH), preeclampsia (PE), placental abruption (PA), fetal growth restriction (FGR), preterm delivery (PTD), and stillbirth (SB) [1]. Based on the probability of complications, primarily DVT, ITs are divided into high-risk, medium-risk, and low-risk categories. High-risk ITs include antithrombin deficiency (ATD) and homozygous mutations of factor V Leiden (FVL) and factor II G20210A (FII). Medium-risk ITs include deficiencies of protein C (PRC) and protein S (PRS), as well as heterozygous FVL and FII mutations. Low-risk ITs include plasminogen activator inhibitor 1 (PAI-1) and methylenetetrahydrofolate reductase (MTHFR) mutations [2].

Inherited thrombophilia is among the most prominent genetic predispositions for thromboembolic events and adverse pregnancy outcomes. Their prevalence varies worldwide depending on the population and specific mutations, with factor V Leiden (FVL) heterozygosity present in about 3–7% of Caucasians and prothrombin G20210A mutation in about 2% of the general population [3,4,5]. In Serbia, FVL is the most common inherited thrombophilia, found in 5–8% of the general population [3,6,7]. The PAI-1 4G/5G polymorphism, another important genetic risk factor for reduced fibrinolysis, may increase the risk of thrombosis and pregnancy complications; the PAI-1 4G/4G genotype is present in 20–25% of healthy individuals in Serbia and even more frequently among women with pregnancy complications [8]. In addition, the prothrombin G20210A (FII) mutation, although less common than FVL, has been reported at an approximately 2–3% prevalence in the Serbian general population, which is consistent with rates observed across Central and Eastern Europe [3,7]. Among women with adverse pregnancy outcomes or thromboembolic events, the frequency of the FII mutation may be somewhat higher, reflecting its clinical significance in this group. Other polymorphisms, such as MTHFR C677T, are also common in the Balkans and may contribute to the overall burden of IT-related pregnancy complications [8]. Given their significant epidemiological value, early detection of these mutations is essential for risk assessment and management, especially in women with previous adverse outcomes. These epidemiological differences emphasize the importance of regional data in guiding both risk assessment and clinical management.

The etiopathogenesis of pregnancy complications (PCs) in patients with IT is not completely understood. The basis of pregnancy complications (PCs) is believed to be pathological placentation caused by trophoblast invasion of spiral arteries. Additionally, hypercoagulability and secondary thrombosis may lead to placental vasculopathy, resulting in impaired perfusion of the intervillous space and subsequent infarctions. These changes may result in pregnancy complications (PCs) [9].

Guidelines for thrombophilia testing include only conditional recommendations, and testing patterns vary widely among different centers [2,10]. Before pregnancy, testing is recommended for patients who have had two or more spontaneous abortions before the 12th week of gestation, one or more stillbirths of unknown etiology in a morphologically healthy fetus, or one or more deliveries before the 34th week caused by preeclampsia, eclampsia, intrauterine fetal growth restriction, or placental abruption, or if they have had a thromboembolic complication [10].

Limited evidence is available to guide the screening and management of these conditions in pregnancy. In the diagnosis of inherited thrombophilia, the necessary tests are activated protein C resistance (APCR), homocysteine level, protein C activity, protein S activity, antithrombin III (ATIII) activity, and prothrombin G20210A and PAI-1 mutation. Confirmatory tests include factor V Leiden (FV-Leiden PCR), antithrombin III (ATIII), pregnancy complications, and protein S (PS) antigen (ELISA) [11,12].

There are many recommendations for anticoagulant use during pregnancy, but many of them are not precise or well-documented [13,14,15]. The potential negative effects of anticoagulant therapy, whether due to failure to administer it or its overuse in patients with IT, remain a topic of debate [16]. However, the effectiveness of anticoagulant therapy in preventing pregnancy complications in pregnant women with IT during pregnancy remains uncertain. The decision to initiate therapy in patients with confirmed IT requires consideration of various risk factors, such as age, obesity, prolonged bed rest during pregnancy, multiple pregnancies, varicose veins, infections, and family history [17]. During pregnancy, low-molecular-weight heparin (LMWH) can be safely used in prophylactic or therapeutic doses, provided that all the aforementioned risk factors are taken into account with regular monitoring of D-dimer anti-Xa levels [18].

The study had two primary goals: to assess whether specific types of IT are associated with distinct pregnancy complications; and to evaluate whether the incidence of pregnancy complications differs between patients with confirmed inherited thrombophilia who received low-molecular-weight heparin (LMWH) therapy and those who did not. The fact that we know that the patient has thrombophilia before a pregnancy should lead to increased awareness and timely prevention of pregnancy complications.

## 2. Materials and Methods

### 2.1. Study Design and Setting

We conducted a retrospective case–control study including 552 participants, divided into two groups: 276 pregnant women diagnosed with inherited thrombophilia (IT) and with at least one prior pregnancy and 276 healthy pregnant women with at least one previous uncomplicated pregnancy and no known risk factors for IT, serving as the control group. The IT group was further subdivided into two cohorts: 108 patients who received low-molecular-weight heparin (LMWH) therapy from the first trimester until 14 days postpartum and 168 patients who did not receive LMWH during pregnancy.

All participants were managed at the University Clinic for Gynecology and Obstetrics “Narodni Front” in Belgrade, Serbia. The study was conducted in accordance with the principles of the Declaration of Helsinki and approved by the Ethics Committee of the clinic (protocol number 22008/2025, approval date: 25 March 2025).

### 2.2. Sampling and Sample Size

The study sample comprised 276 pregnant women with at least one prior pregnancy complicated by an adverse obstetric outcome. All participants in the study group had undergone thrombophilia testing and were diagnosed with inherited thrombophilia (IT) prior to the current pregnancy. For comparison, the control group included 276 healthy pregnant women, selected based on the absence of personal or family history of thrombophilia or other established risk factors. This group was used exclusively for evaluating pregnancy outcomes such as gestational age at delivery, birth weight, Apgar scores, and mode of delivery in order to compare clinical outcomes between IT patients and a low-risk obstetric population. In the IT group, D-dimer levels were systematically measured in all patients during the first trimester, as part of risk stratification and treatment planning. In the control group, D-dimer testing was performed only in patients who conceived via in vitro fertilization (IVF), where the test was included in standard preconception screening protocols. No thrombophilia testing or D-dimer measurements were otherwise performed in the control group, as there were no clinical indications or known risk factors that would justify such evaluation. Of the 276 women in the study group, 30 experienced pregnancy loss before 12 weeks of gestation. As neonatal outcomes such as birth weight, Apgar scores, and gestational age at delivery could not be assessed in these cases, the comparative analysis of perinatal parameters was conducted on the remaining 246 pregnancies that progressed beyond the first trimester.

### 2.3. Inclusion and Exclusion Criteria

Inclusion Criteria: The study group included pregnant women diagnosed with inherited thrombophilia (IT) prior to the current pregnancy. The diagnosis was established based on one or more of the following: A personal history of deep vein thrombosis (DVT), prior pregnancy complications (PCs), or a confirmed positive family history (i.e., first-degree relatives with inherited thrombophilia).

Exclusion Criteria: Of the 283 patients initially screened for inclusion in the IT group, 7 were excluded due to low subgroup numbers: 3 patients with homozygous factor V Leiden (FVL) mutation, 1 with homozygous prothrombin G20210A mutation, and 3 with protein S (PSD) or protein C deficiency (PCD). Additionally, twin pregnancies and cases of acquired thrombophilia were excluded from the analysis.

### 2.4. Patient Characteristics

In the study group, the most frequently observed form of inherited thrombophilia (IT) was homozygous PAI-1 4G/4G mutation (55.1%), followed by heterozygous factor V Leiden (FVL) (27.5%), heterozygous prothrombin G20210A (FII) mutation (10.1%), and homozygous MTHFR C677T mutation (7.2%) (Figure 1). All participants had a history of at least one prior pregnancy complicated by an adverse obstetric outcome, including preeclampsia (PE), gestational hypertension (GH), preterm delivery (PTD), stillbirth (SB), deep vein thrombosis (DVT), fetal growth restriction (FGR), HELLP syndrome, or premature rupture of membranes (PROM). Some patients were tested for thrombophilia based on a positive family history of DVT in first-degree relatives, in addition to their own adverse pregnancy history. Nulliparous patients were included based on a history of two or more consecutive first-trimester pregnancy losses (prior to 12 weeks of gestation). The diagnosis of IT was established prior to the current pregnancy; thus, none of the patients received LMWH therapy in earlier pregnancies, except for those with a documented history of DVT. Although the MTHFR C677T mutation is common in the general population and is no longer considered clinically significant per the latest guidelines from the American College of Obstetricians and Gynecologists (ACOG) and the Royal College of Obstetricians and Gynecologists (RCOG), it was included in our analysis due to its co-occurrence with prior adverse pregnancy outcomes [18,19]. All pregnancies included in the study were singletons, with confirmed gestational age and no evidence of congenital anomalies or intrauterine infection.

### 2.5. Selective Therapy: LMWH

#### 2.5.1. Initiation of Therapy

LMWH therapy was initiated during the first trimester of the current pregnancy, following ultrasound confirmation of an intrauterine gestation with a documented fetal heartbeat. The decision to initiate therapy was individualized and based on multiple clinical factors, including the specific subtype of inherited thrombophilia (IT), obstetric history, maternal age, body mass index (BMI), and first-trimester D-dimer levels. LMWH was administered to 108 patients (39.13%) in the IT group. Among these, factor V Leiden (FVL) mutation was the most common indication for treatment, observed in approximately 75% of treated cases (Figure 1).

#### 2.5.2. The LMWH Dosage

Therapy was administered in either prophylactic or therapeutic doses. In addition to predefined initiation criteria, treatment decisions were further individualized based on the thrombogenic potential of the specific inherited thrombophilia (IT) mutation and the severity of previous pregnancy complications (PCs), such as deep vein thrombosis (DVT), placental abruption (PA), severe early-onset preeclampsia (PE), or early fetal growth restriction (FGR). Other contributing factors included maternal age over 40 years, elevated body mass index (BMI) > 28 kg/m^3^, and D-dimer levels (>1.42 mg/L FEU) [18].

Criteria for prophylactic LMWH therapy included: Advanced maternal age, confirmed moderate-risk thrombophilia (e.g., heterozygous factor V Leiden or prothrombin G20210A mutations), or low-risk IT (e.g., PAI-1 4G/4G) accompanied by a history of PE, HELLP syndrome, PA, early FGR, or PTD in a previous pregnancy. Elevated first-trimester D-dimer values (>1.42 mg/L FEU) and a positive family history of venous thromboembolism (in first-degree relatives) were also considered as supportive risk factors (subindications) contributing to the decision to initiate prophylactic treatment. In one patient with MTHFR mutation, a prophylactic dose was started during IVF preparation and continued throughout pregnancy due to elevated D-dimer levels and advanced maternal age.

Therapeutic LMWH dosing was reserved for patients with high-risk thrombophilia (i.e., heterozygous FVL or FII mutations) in the presence of additional clinical risk factors, including a personal history of DVT or stillbirth, persistently elevated D-dimer levels in the first trimester, or BMI > 28 kg/m^2^.

LMWH was not administered to patients under 35 years of age with low-risk thrombophilia (e.g., homozygous PAI-1 or MTHFR) who had no additional risk factors and presented with normal D-dimer values (<0.58 mg/L FEU) and BMI < 24 kg/m^2^. Patients with isolated histories of premature rupture of membranes (PROM), oligohydramnios (OH), or recurrent pregnancy loss (RPL) were likewise managed without anticoagulation.

Of the 108 patients who received LMWH therapy, 81 were treated with prophylactic doses, including nadroparin (Fraxiparine 2850 IU once daily), enoxaparin (Clexane 40 mg once daily), or dalteparin (Fragmin 5000 IU once daily). Therapeutic LMWH dosing was used in 27 patients and included weight-based regimens: nadroparin (86 IU/kg every 12 h), dalteparin (100 IU/kg every 12 h), or enoxaparin (1 mg/kg every 12 h).

#### 2.5.3. Monitoring and Safety

Patients receiving LMWH therapy underwent routine antenatal monitoring, which included clinical assessments as well as laboratory evaluations of D-dimer levels and platelet counts throughout pregnancy. Anti-Xa activity was not routinely measured, in accordance with standard low-risk monitoring protocols for prophylactic dosing. The therapy was generally well-tolerated, and no maternal bleeding complications or adverse drug reactions were observed in the LMWH-treated group.

### 2.6. Data Collection

Comprehensive clinical data were collected for all participants, including maternal age, parity, and mode of conception (spontaneous vs. in vitro fertilization/embryo transfer (IVF/ET)). A detailed obstetric history was obtained, covering prior pregnancy complications (PCs) such as deep vein thrombosis (DVT), miscarriage, preterm delivery (PTD), gestational hypertension (GH), preeclampsia (PE), fetal growth restriction (FGR), placental abruption (PA), stillbirth (SB), oligohydramnios (OA), and premature rupture of membranes (PROM). In addition, first-trimester D-dimer levels and maternal body mass index (BMI) were recorded, as well as the presence of a positive family history of venous thromboembolism (VTE) in first-degree relatives. Perinatal outcomes assessed in the current pregnancy included gestational age at delivery, mode of delivery (vaginal vs. cesarean section (CS)), neonatal birth weight, and 1 and 5 min Apgar scores. In the study group, data were collected both on prior pregnancies without LMWH therapy and on the current pregnancy following the diagnosis of inherited thrombophilia (IT), with selective LMWH administration based on predefined criteria.

Pregnancy complications were retrospectively analyzed in two distinct periods: prior pregnancies, before the diagnosis of IT and without LMWH therapy, and the current pregnancy, after the diagnosis of IT, with LMWH selectively administered. In the current pregnancy, patients were categorized into two subgroups: IT patients who received LMWH therapy and IT patients who did not receive LMWH therapy. The effect of IT on pregnancy complications was analyzed by comparing the incidence of pregnancy complication, before and after diagnosis, regardless of LMWH use. Furthermore, genotype–phenotype correlations were explored by comparing the frequency of specific complications according to the thrombophilia mutation type (FVL, FII G20210A, PAI-1, or MTHFR).

As this was a retrospective study, a cohort of 276 low-risk pregnancies was deliberately selected to serve as a reference group. The primary objective of this comparison was to evaluate whether pregnancy and neonatal outcomes, such as gestational age at delivery, neonatal birth weight, and Apgar scores and cesarean section (CS) rate, differed between women with IT and those from the general obstetric population. In addition to perinatal outcomes, maternal body mass index (BMI) was also compared between groups, in order to explore its potential contribution to pregnancy risk and outcomes in women with IT. This comparative analysis aimed to determine whether IT itself, independent of other risk factors, was associated with an increased risk of adverse neonatal outcomes.

A key objective of the study was to evaluate whether favorable pregnancy outcomes could be achieved without LMWH in women with moderate- or low-risk thrombophilia who had a history of adverse outcomes in prior pregnancies but lacked additional risk factors (e.g., elevated D-dimer or abnormal BMI) in the current pregnancy. The primary outcome was the frequency of major pregnancy complications, including preterm delivery, fetal growth restriction, gestational hypertension, and recurrent pregnancy loss, in the same women before and after inherited thrombophilia diagnosis. Complications were compared between previous pregnancies (before diagnosis, untreated) and current pregnancies (after diagnosis, managed with or without LMWH). The effect of LMWH therapy was assessed by comparing the frequency of pregnancy complications in the current pregnancy between women who received LMWH and those who did not. This subgroup analysis aimed to inform a more selective, risk-adapted approach to LMWH use in pregnancy. Such a strategy may help reduce overtreatment and optimize resource allocation, while maintaining safety for patients unlikely to benefit from anticoagulation. Lastly, pregnancy outcomes were analyzed separately for FVL carriers to evaluate whether LMWH administration was associated with a reduction in adverse events, acknowledging the limited power of subgroup analyses with small sample sizes.

### 2.7. Statistical Analysis

Descriptive statistics were used to summarize patient characteristics and outcome measures. The normality of data distribution was assessed using the Kolmogorov–Smirnov test. Depending on data distribution and variable type, the following tests were applied for group comparisons: Student’s *t*-test, Wilcoxon signed-rank test, Mann–Whitney U test, and Fisher’s exact test. A two-tailed *p* value of <0.05 was considered statistically significant. All statistical analyses were conducted using Microsoft Excel and MedCalc software (version 23.0.9; MedCalc Software Ltd., Ostend, Belgium).

## 3. Results

Statistically significant differences were observed between the inherited thrombophilia (IT) and control groups across several maternal and pregnancy-related characteristics. Women in the IT group were significantly younger (34.1 vs. 39.2 years, *p* < 0.001), had a higher BMI (24.3 vs. 23.3, *p* < 0.001), and more frequently conceived via IVF/ET (18.1% vs. 12%, *p* = 0.043). First-trimester D-dimer levels were markedly elevated in the IT group (1.21 vs. 0.04 mg/L, *p* < 0.001), and the mean gestational age at delivery was significantly lower (37.4 vs. 32.0 weeks, *p* < 0.001). These findings highlight a higher baseline risk profile and a more complex obstetric course in women with inherited thrombophilia compared to the control group, which represents a low-risk, healthy obstetric population. These findings suggest that, even with antenatal monitoring and selective LMWH therapy, inherited thrombophilia remains associated with higher risk for suboptimal pregnancy and neonatal outcomes. This supports the clinical need for early identification and risk-based management of affected pregnancies (Table 1).

This table presents the frequency of pregnancy complications among women with inherited thrombophilia during previous and current pregnancies (Table 2). It aims to evaluate the potential impact of diagnosis, clinical awareness, and interventions such as low-molecular-weight heparin (LMWH) therapy and closer antenatal monitoring. Across most complications—including premature rupture of membranes (PROM) (15.6% to 3.3%), OA (18.1% to 4.7%), PTD (14.1% to 4.4%), RPL (19.2% to 10.9%), and gestational hypertension (GH) (9.1% to 6.5%)—a reduction in frequency was observed during the current pregnancy. Although these differences did not reach statistical significance (*p* > 0.05), the magnitude of reduction is clinically meaningful. The greatest decreases were seen in premature rupture of membranes (PROM) and preterm delivery (PTD), which are often associated with thrombo-inflammatory placental processes. Interestingly, fetal growth restriction (FGR) showed a decrease (10.5% to 6.9%) with an odds ratio (OR = 0.36) and relative risk (RR = 0.38) below 1, suggesting a potential protective trend—possibly related to targeted therapy and surveillance. Similarly, preeclampsia (PE) and gestational hypertension (GH) rates declined in the current pregnancy, though not significantly. However, due to very low case numbers, complications such as placental abruption (PA), hemolysis (HELLP), stillbirth (SB), and deep vein thrombosis (DVT) yield wide confidence intervals or undefined ORs/RRs (reported as 0.0), limiting interpretability. These findings support a trend toward improved obstetric outcomes in the current pregnancy among women with inherited thrombophilia, likely due to earlier diagnosis, risk-based stratification, and individualized management.

Among women with inherited thrombophilia, the PAI-1 4G/4G mutation was significantly associated with gestational hypertension (GH) (*p* = 0.000), suggesting a potential role of impaired fibrinolysis in pregnancy-related hypertensive disorders. The factor V Leiden (FVL) group also showed a significant association with GH (*p* = 0.01) and a trend toward increased fetal growth restriction (FGR) (*p* = 0.06). No statistically significant associations were found for the prothrombin G20210A (FII) or MTHFR groups, though small sample sizes may limit power. These findings support genotype-specific risk profiling in the management of thrombophilic pregnancies (Table 3).

A statistically significant reduction in the incidence of fetal growth restriction (FGR) was found among women treated with LMWH compared to untreated patients (*p* = 0.007). Preterm delivery (PTD) showed a trend toward lower rates in the LMWH group; this was statistically significant in the one-sided Fisher’s exact test (*p* = 0.042), although not by chi-square (*p* = 0.089). For other complications (PROM, OA, GH, PA, HELLP, SB, DVT, and early pregnancy loss), no significant differences were observed between groups (*p* > 0.05). These results support the potential benefit of LMWH in reducing FGR and, to a lesser extent, PTD (Table 4).

The Table 5 presents the incidence of pregnancy complications among women who received LMWH therapy versus those who did not, along with key clinical effect measures: Absolute risk reduction (ARR), relative risk reduction (RRR), and number needed to treat (NNT). A positive ARR and a low NNT suggest that LMWH therapy may be effective in preventing the respective complications. In contrast, a negative ARR indicates that the complication occurred more frequently in the LMWH group. However, this does not necessarily imply treatment failure; rather, it may reflect that LMWH recipients had a higher baseline risk and were selectively treated based on clinical severity or adverse obstetric history. Among all outcomes, oligohydramnios (OA) demonstrated the most favorable response to LMWH, with a positive ARR and an NNT of 94. Conversely, complications such as fetal growth restriction (FGR), preterm delivery (PTD), and gestational hypertension (GH) showed negative ARR values—likely reflecting a treatment bias, where LMWH was administered to patients with greater underlying risk. These results underscore the importance of interpreting clinical effect measures within the context of patient selection and baseline characteristics, rather than assuming causal relationships from unadjusted comparisons.

This forest plot (Figure 2) displays the odds ratios (OR) and their 95% confidence intervals (CIs) for various pregnancy complications, comparing outcomes in women treated with LMWH to those who were not. In this analysis, although the ORs for several complications were either above or below 1, most confidence intervals crossed the line of no effect, indicating that these findings were not statistically significant. However, fetal growth restriction (FGR) showed a trend toward reduction in the LMWH group, consistent with previously reported potential benefits of LMWH in improving placental perfusion.

Low-molecular-weight heparin (LMWH) therapy in patients with the factor V Leiden (FVL) mutation was associated with a statistically significant reduction in fetal growth restriction (FGR; *p* = 0.0005), preeclampsia (PE; *p* = 0.013), gestational hypertension (GH; *p* = 0.012), and recurrent pregnancy loss (RPL; *p* = 0.012). These findings suggest a potential benefit of LMWH in mitigating thrombotic and placenta-mediated complications in this high-risk subgroup. Conversely, no significant differences were observed between treated and untreated groups for complications such as preterm delivery (PTD), preterm premature rupture of membranes (PPROM), or HELLP syndrome, indicating that the therapeutic effect of LMWH may be selective and condition specific (Table 6).

This logistic regression model evaluated the independent effect of LMWH therapy and selected clinical parameters on the occurrence of any pregnancy complication in the current pregnancy (Table 7). LMWH therapy was not significantly associated with a reduced or increased risk of complications (OR: 1.071; 95% CI: 0.894–1.281; *p* = 0.462), suggesting no statistically meaningful benefit for the overall complication rate in this cohort. Likewise, maternal age, BMI, IVF conception, and D-dimer levels in the first trimester did not significantly predict the occurrence of complications. These results emphasize the need for individualized risk assessment, as LMWH may provide benefit for specific outcomes rather than universally reducing complication risk.

## 4. Discussion

Our findings suggest that low-molecular-weight heparin (LMWH) therapy does not provide a uniform protective effect across all pregnancy complications (PCs) in women with inherited thrombophilia (IT) but rather offers selective benefit in genetically and clinically defined high-risk subgroups. In our cohort, statistically significant reductions were observed in the incidence of fetal growth restriction (FGR), gestational hypertension (GH), preeclampsia (PE), and recurrent pregnancy loss (RPL) among factor V Leiden (FVL) carriers who received LMWH therapy. These outcomes support existing evidence indicating that FVL is a highly thrombogenic mutation with strong associations with placenta-mediated complications [3,10,20]. In contrast, no statistically significant benefit of LMWH was observed in women carrying prothrombin G20210A (FII) or MTHFR mutations, likely due to both smaller subgroup sizes and lower baseline risk [21,22]. Although a general trend toward reduced pregnancy complications was observed in the current pregnancy compared to previous ones—regardless of LMWH use, this likely reflects the impact of improved antenatal surveillance, early risk identification, and individualized care protocols rather than a universal pharmacologic effect [10,11]. Furthermore, multivariable logistic regression did not identify LMWH therapy as an independent predictor of reduced overall pregnancy complications, suggesting that its clinical benefit is not absolute but conditional on underlying genetic, clinical, and biochemical risk factors. These results reinforce the importance of a risk-adapted therapeutic approach, consistent with current guidelines [18,19]. For women with low- to moderate-risk mutations (e.g., isolated heterozygous MTHFR or PAI-1 4G/4G), and no additional risk factors such as elevated body mass index (BMI), increased D-dimer levels, or adverse obstetric history, favorable outcomes were often achieved through close prenatal surveillance alone. While LMWH appears beneficial in reducing specific complications, particularly in high-risk cases, its indiscriminate use raises concerns regarding cost-effectiveness, overtreatment, and therapy-related risks such as bleeding or thrombocytopenia. The number needed to treat (NNT) analysis supports this selective strategy: beneficial NNTs were observed for complications like oligohydramnios (94), placental abruption (378), and stillbirth (168), while negative absolute risk reductions (ARRs) for complications such as fetal growth restriction (FGR) and preterm delivery (PTD) likely reflect selection bias, with therapy reserved for those with higher baseline risk (Table 5). In conclusion, LMWH therapy should not be routinely administered to all women with IT. Instead, its application should be based on individualized risk assessment, incorporating thrombophilia subtypes, prior obstetric history, and additional clinical markers. This approach optimizes maternal and fetal outcomes while minimizing unnecessary treatment. In our study, three types of low-molecular-weight heparins (LMWHs), nadroparin (Fraxiparine), enoxaparin (Clexane), and Dalteparin (Fragmin), were administered either prophylactically or therapeutically based on thrombophilia subtype, prior pregnancy complications, and individualized risk assessment. Although all LMWHs share a common mechanism of action by selectively inhibiting factor Xa, minor pharmacokinetic differences exist that may influence clinical practice [23]. However, the clinical relevance of these differences during pregnancy remains limited, as direct comparisons have not shown consistent superiority of one agent over another in obstetric outcomes [24]. The choice of LMWH formulation in our cohort was primarily guided by local availability and clinical familiarity rather than strong evidence favoring one agent. Given the lack of head-to-head comparative trials of LMWHs in pregnancy, further prospective studies are needed to clarify whether certain therapies may offer superior maternal–fetal safety or efficacy profiles in specific subgroups of thrombophilia patients [24].

Multivariate analysis revealed no significant association between LMWH use and overall complication risk, underscoring the need for individualized therapy rather than routine use. Current guidelines also endorse a personalized, risk-adapted approach, integrating thrombophilia subtype, clinical history, and biomarkers to guide treatment decisions. Our findings support this model and highlight the need for further prospective studies to optimize thromboprophylaxis in pregnancy. Our findings confirm that low-molecular-weight heparin (LMWH) therapy does not uniformly reduce the overall incidence of pregnancy complications in women with inherited thrombophilia (IT). However, subgroup analysis and absolute risk metrics suggest selective benefits in defined clinical contexts. Positive absolute risk reduction (ARR) values and low numbers needed to treat (NNTs) were observed for oligohydramnios (NNT = 94), placental abruption (NNT = 378), and stillbirth (NNT = 168), indicating potential efficacy in preventing these outcomes. In contrast, negative ARR values for complications like fetal growth restriction (FGR) and gestational hypertension (GH) likely reflect risk selection, where LMWH was preferentially prescribed to higher-risk patients. In our cohort, LMWH-treated factor V Leiden carriers exhibited significant reductions in fetal growth restriction (FGR), gestational hypertension (GH), preeclampsia (PE), and recurrent pregnancy loss (RPL), supporting its role in improving placental perfusion and mitigating thrombotic complications. These findings are consistent with prior research suggesting that LMWH enhances uteroplacental circulation and reduces inflammation and oxidative stress associated with placental dysfunction [18,19,22,25]. Multivariate analysis revealed no statistically significant association between LMWH use and overall complication risk (OR: 1.071; 95% CI: 0.894–1.281; *p* = 0.462), reinforcing the heterogeneity of thrombophilia-associated outcomes and the limitations of a universal approach. This supports guideline-based strategies that advocate for individualized use of LMWH based on genotype, clinical history, and coexisting risk factors [18,19,26,27]. Future prospective studies with stratified subgroups are needed to refine the therapeutic indications for LMWH in pregnancy.

This study confirms a well-established association between inherited thrombophilia (IT) and adverse pregnancy outcomes [9,10]. In our cohort, women with IT were significantly older and had higher BMI values than healthy controls, both of which are recognized contributors to hypertensive disorders, fetal growth restriction (FGR), and preterm delivery (PTD). Elevated first-trimester D-dimer levels (1.21 mg/L vs. 0.39 mg/L) further support the presence of a hypercoagulable state and underline the pathophysiologic link between IT and placental dysfunction [9,10]. These maternal and biochemical factors likely contributed to the significantly lower birth weight, earlier gestational age at delivery, and higher cesarean section rate observed in the IT group. The increased rate of cesarean section may reflect a more cautious clinical approach due to maternal risk factors and prior adverse obstetric history [11,12]. While more IVF pregnancies were seen in the IT group, this likely reflects preconception screening patterns rather than a direct causal link [17,28].

Inherited thrombophilia (IT), including factor V Leiden (FVL), prothrombin G20210A (FII), and PAI-1 4G/5G polymorphisms, is known to predispose individuals to thrombotic events and has been variably associated with adverse pregnancy outcomes [4,10,18]. In our cohort, the most prevalent mutations were homozygous PAI-1 4G/4G (55.1%) and heterozygous FVL (27.5%), consistent with previous reports in European Caucasian populations [29,30]. In Serbia, FVL has been identified as a common risk factor for venous thromboembolism (VTE) with a significantly higher prevalence among deep vein thrombosis (DVT) and preeclampsia (PE) patients compared to controls [3,29]. Furthermore, prior studies have implicated PAI-1 in reproductive complications, particularly in the context of combined thrombophilic genotypes [4,21,31]. Comparative analysis of previous and current pregnancies revealed a clinically meaningful reduction in the incidence of several complications. Preterm premature rupture of membranes (PPROM) decreased from 15.6% to 3.3% and preterm delivery (PTD) from 14.1% to 4.4%, consistent with the hypothesis that early identification and individualized care, potentially including LMWH, may reduce thrombo-inflammatory placental injury. While not statistically significant (PPROM: *p* = 0.927; PTD: *p* = 0.523), these findings suggest a potential therapeutic trend. Notably, fetal growth restriction (FGR) decreased from 10.5% to 6.9%, with an OR of 0.36 and RR of 0.38, indicating a possible protective effect. Similar downward trends were observed in gestational hypertension (GH) and preeclampsia (PE), further supporting the value of targeted surveillance and prophylaxis. However, due to low event counts, conclusions regarding rarer outcomes such as placental abruption (PA), HELLP SY, stillbirth (SB), and deep vein thrombosis (DVT) remain limited. Overall, our findings emphasize the importance of structured prenatal care and genotype-informed management strategies in improving obstetric outcomes among women with IT.

Analysis by thrombophilia subtype revealed genotype-specific patterns of complications. Among factor V Leiden (FVL) carriers, gestational hypertension (GH) was significantly more frequent (*p* = 0.01), and a trend toward increased fetal growth restriction (FGR) (*p* = 0.06) was noted, suggesting a possible link to placental dysfunction [3,18]. Although not statistically significant, the observed higher rates of preeclampsia (PE) in MTHFR carriers (10%) may reflect underlying risk but are limited by small sample size and ongoing debate about its clinical impact. No significant associations were found in the FII group, likely due to low statistical power (N = 28). The PAI-1 4G/4G group had the highest overall complication rate (11.9%), supporting evidence that impaired fibrinolysis may contribute to hypertensive disorders and pregnancy loss [32]. Prior studies and meta-analyses have linked PAI-1 mutations, particularly when combined with other thrombophilic factors, to recurrent pregnancy loss [21,22,33]. These findings emphasize the need for genotype-based risk stratification. In particular, homozygous PAI-1 and high-risk FVL carriers may benefit from prophylactic LMWH and enhanced monitoring, especially in the presence of additional risk factors like obesity or prior obstetric complications. Collectively, this supports a multifactorial pathophysiological model of IT, involving both thrombotic and non-thrombotic mechanisms, and reinforces the importance of early identification and individualized care in this population [20,24,34].

This study has some limitations that should be acknowledged. First, its retrospective design may introduce recall bias and limit causal inferences. Although predefined criteria were used to guide LMWH therapy, treatment decisions were also influenced by individual clinical judgment, resulting in heterogeneity in therapy allocation. Anti-Xa levels were not routinely measured, which may have limited the ability to verify therapeutic adequacy, particularly in higher-risk patients or those with elevated BMI. The use of three different LMWH agents (nadroparin, enoxaparin, and dalteparin) without randomization or stratification further limits conclusions about comparative efficacy. Additionally, sample sizes for certain thrombophilia subtypes, especially untreated FII mutation carriers, were small, reducing statistical power in subgroup analyses. Pregnancy complications were assessed based on clinical records, which may be subject to documentation variability or misclassification, especially for earlier pregnancies. Biochemical markers of placental function, such as pregnancy-associated plasma protein A (PAPP-A), placental growth factor (PlGF), and soluble fms-like tyrosine kinase-1 (sFlt-1), were not routinely measured, which limits insight into underlying pathophysiological mechanisms. Furthermore, the study did not explore the psychological impact of thrombophilia diagnosis or treatment, nor were patient-reported outcomes assessed, despite their relevance to treatment adherence and perceived quality of care. Finally, while obstetric outcomes were carefully monitored, the study did not assess long-term neonatal outcomes or maternal quality of life, both of which are important considerations in evaluating the full impact of thromboprophylaxis during pregnancy. These limitations highlight the need for larger, multicenter prospective studies with standardized protocols and broader outcome assessments to more definitively determine the role of LMWH in the management of inherited thrombophilia during pregnancy. In addition, the enrollment periods for the thrombophilia and control groups were not fully synchronized, which may introduce temporal bias and limit comparability of obstetric management practices over time.

## 5. Conclusions

Our study demonstrates that low-molecular-weight heparin (LMWH) prophylaxis confers significant benefit primarily in high-risk subgroups of women with inherited thrombophilia (IT)—specifically those with a history of severe pregnancy complications or carriers of highly thrombogenic mutations such as factor V Leiden (FVL). In contrast, LMWH therapy provided limited or no benefit in women with low-risk genotypes and no additional risk factors, where structured prenatal monitoring without routine anticoagulation may be sufficient. We confirm that inherited thrombophilia is associated with an increased risk of adverse pregnancy outcomes, including earlier delivery, lower neonatal birth weight, and higher rates of cesarean section. In our cohort, homozygous PAI-1 and heterozygous FVL mutations were the most common, while women with IT tended to be older and have higher BMI and elevated first-trimester D-dimer levels.

LMWH prophylaxis should be reserved for high-risk women with inherited thrombophilia, whereas routine use in all IT pregnancies is not justified and may lead to unnecessary risks and healthcare costs. Early thrombophilia screening, careful risk stratification, and individualized management are essential to optimize maternal and perinatal outcomes. Further prospective studies are warranted to refine therapeutic strategies in this population.

## Figures and Tables

**Figure 1 medicina-61-01131-f001:**
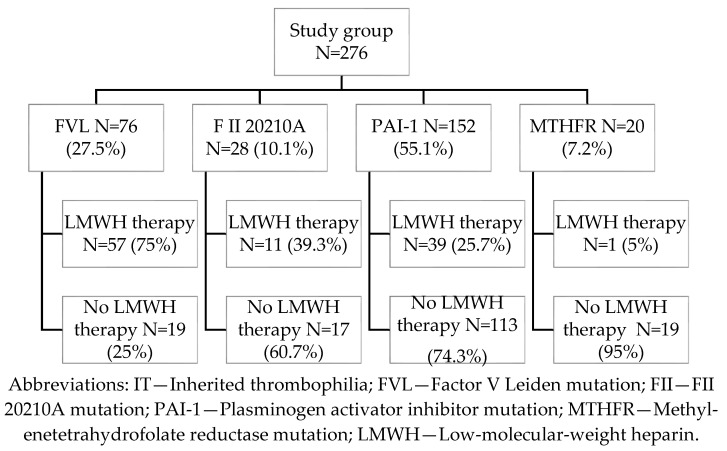
Flow chart of incidence of IT and treatment with LMWH in the study group.

**Figure 2 medicina-61-01131-f002:**
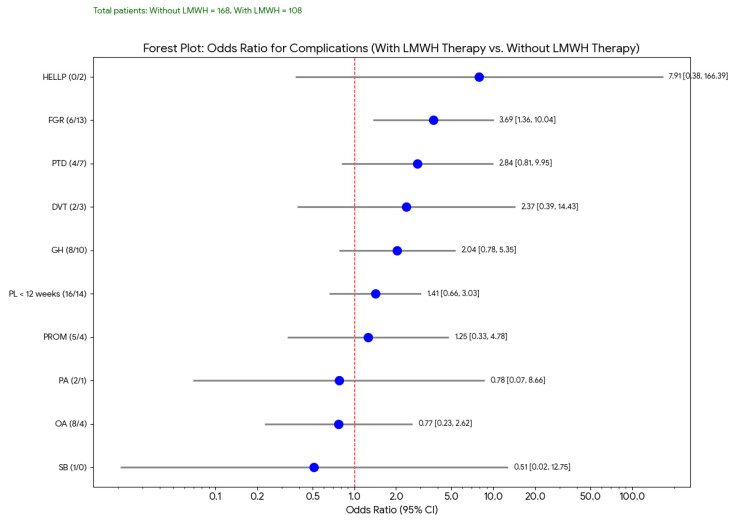
Effect of Low-Molecular-Weight Heparin Therapy on Major Pregnancy Complications: Forest Plot Analysis.

**Table 1 medicina-61-01131-t001:** Baseline maternal characteristics and pregnancy outcomes in women with inherited thrombophilia and healthy controls.

Group Characteristics	Inherited Thrombophilia N = 276	Control Group N = 276	*p* Value
Maternal age BMI	34.1 ^+/−^ 3.9 * 24.34 ^+/−^ 3.16 *	32.1 ^+/−^ 4.1 23.2 ^+/−^ 3.11	<0.05 <0.05
IVF/ET	50 (18.12%) *	33 (12%)	<0.05
Parity n (%)			<0.05
1	50 (18.12%)	137 (49.64%)	
2	168 (60.87%)	103 (37.32%)	
>3 D-dimer	58 (21.01%) 1.15 ^+/−^ 1.28 mg/L	36 (13.04%) Not tested	
Family history DVT	15 (5.43%)	0	
Gestational age at delivery (weeks)	37.43 ^+/−^ 1.83 *	39.23 ^+/−^ 9.2	<0.05
Neonatal body weight (g)	3102 ^+/−^ 681 *	3442 ^+/−^ 21	<0.05
1 min Apgar score	8.46 ^+/−^ 1.14 *	9.08 ^+/−^ 1.41	<0.05
5 min Apgar score	9.43 ^+/−^ 1.15 *	9.88 ^+/−^ 1.12	<0.05
Cesarean section n (%)	87 (31.52%) *	27 (9.8%)	<0.05

* *p* < 0.05; Abbreviations: IVF/ET—In Vitro Fertilization/Embryo Transfer. DVT—Deep Vein Thrombosis. D-dimer levels in the control group were not routinely tested; values were available only for 33 women who conceived via IVF/ET. Of the 276 patients with inherited thrombophilia, 30 experienced pregnancy loss before 12 weeks’ gestation. Analyses of neonatal outcomes (birth weight, Apgar scores, gestational age at delivery, and mode of delivery) were conducted on the remaining 246 patients with viable pregnancies beyond the first trimester.

**Table 2 medicina-61-01131-t002:** Frequency of pregnancy complications in previous versus current pregnancy among women with inherited thrombophilia.

Pregnancy Complication	IT Previous Pregnancies N = 276 N (%)	IT Current Pregnancy N = 276 N (%)	Pearson Chi-Square/ Asymptotic Significance *p*	OR	RR
PROM	43 (15.58)	9 (3.3)	0.9271	1.57	1.55
OA	50 (18.12)	13 (4.71)	0.1135	3.03	2.82
PTD	40 (14.13)	12 (4.35)	0.5234	2.05	1.97
FGR	35 (10.51)	19 (6.9)	0.5158	0.36	0.38
PA	11 (1.81)	5 (1.1)	1.0	0.0	0.0
HELLP	8 (2.54)	2 (0.7)	1.0	0.0	0.0
PE	16 (5.07)	11 (3.99)	1.0	1.67	1.62
SB	3 (0.72)	3 (0.72)	1.0	0.0	0.0
GH	26 (9.06)	18 (6.5)	0.8703	0.55	0.57
RPL	53 (19.2)	30 (10.87)	0.1787	1.789	1.8
DVT	9 (3.26)	5 (1.81)	1.0	0.0	0.0

Abbreviations: OC—Obstetric Complications; PROM—Premature Rupture of Membranes; OA—Oligohydramnios; PTD—Preterm Delivery; FGR—Fetal Growth Restriction; PA—Placental Abruption; HELLP—Hemolysis, Elevated Liver Enzymes, Low Platelet Count; PE—Preeclampsia; SB—Stillbirth; GH—Gestational Hypertension; RPL—Recurrent Pregnancy Loss; DVT—Deep Vein Thrombosis; OR—Odds Ratio; RR—Relative Risk. OR or RR reported as 0.00 indicates undefined values due to zero event count in one of the compared groups.

**Table 3 medicina-61-01131-t003:** Frequency of pregnancy complications during current pregnancies in relation to the type of inherited thrombophilia.

Pregnancy Complication Ongoing Pregnancies	FVL N = 76 (%)	*p* Value Fisher’s	FII N = 28 (%)	*p* Value Fisher’s	PAI-1 N = 152 (%)	*p* Value Fisher’s	MTHFR N = 20 (%)	*p* Value Fisher’s
PROM	5	0.18	1	0.47	3	0.06	0	1.0
OA	4	0.75	1	1.0	6	0.57	1	1.0
PTD	2	0.52	1	1.0	7	1.0	1	0.63
FGR	9	0.06	2	1.0	7	0.15	0	0.37
PA	0	0.32	0	1.0	3	1.0	0	1.0
HELLP	2	0.07	0	1.0	0	0.20	0	1.0
PE	3	1.0	2	0.30	4	0.22	2	0.18
SB	1	1.0	0	1.0	1	0.58	0	1.0
GH	10	0.01 *	4	0.09	0	0.000 *	0	0.60
RPL	10	0.51	2	0.75	17	1.0	1	0.70
DVT	3	0.13	1	0.41	1	0.17	0	1.0
Total	25 (9.1)		9 (3.2)		33 (11.9)		4 (1.45)	

* Statistically significant values (*p* < 0.05). Abbreviations: PC—Pregnancy Complications; FVL—Factor V Leiden Mutation; FII—FII G20210A Mutation; PAI-1—Plasminogen Activator Inhibitor Mutation; MTHFR—Methylenetetrahydrofolate Reductase Mutation; CG—Control Group; PROM—Premature Rupture of Membranes; OA—Oligohydramnios; PTD—Preterm Delivery; FGR—Fetal Growth Restriction; PA—Placental Abruption; HELLP—Hemolysis, Elevated Liver Enzymes, Low Platelet Count; PE—Preeclampsia; SB—Stillbirth; GH—Gestational Hypertension; RPL—Recurrent Pregnancy Loss; DVT—Deep Vein Thrombosis.

**Table 4 medicina-61-01131-t004:** The influence of applied Low-Molecular-Weight Heparin therapy on the development of complications in the ongoing pregnancy.

Pregnancy Complication	Without LMWH Therapy N = 168	With MWH Therapy N = 108	Pearson Chi-Square		Fisher’s Exact Test	
			Value	Asymptotic Significance	Exact Sig (2-Sided)	Exact Sig (1-Sided)
PROM	5	4	0.110 ª	0.740	0.741	0.496
OA	8	4	0.177 ª	0.674	0.770	0.461
PTD	4	7	2.899 ª	0.089	0.116	0.085
PL < 12 weeks	16	14	0.487 ª	0.485	0.429	0.863
GH	8	10	2.181 ª	0.140	0.210	0.111
FGR	6	13	7.350 ª	0.007	0.013	0.007
PA	2	1	0.430 ª	0.836	1.000	0.661
HELLP	0	2	3.134 ª	0.077	0.152	0.152
SB	1	0	0.0 ª	1.0	1.000	0.609
DVT	2	3	0.0 ª	1.0	1.0	0.699
Total	52 (30.95%)	58 (53.70%)				

ª Pearson chi-square test with Yates’ continuity correction. Abbreviations: LMWH—Low-Molecular-Weight Heparin; PROM—Premature Rupture of Membranes; OA—Oligohydramnios; PTD—Preterm Delivery; PL—Pregnancy Loss; GDM—Gestational Diabetes Mellitus; GH—Gestational Hypertension; FGR—Fetal Growth Restriction; PA—Placental Abruption; HELLP—Hemolysis, Elevated Liver Enzymes, Low Platelet Count; SB—Stillbirth; DVT—Deep Vein Thrombosis.

**Table 5 medicina-61-01131-t005:** Clinical Impact of Low-Molecular-Weight Heparin on Pregnancy Complications: Absolute Risk Reduction, Relative Risk Reduction, and Number Needed to Treat.

Complication	Risk Without LMWH	Risk with LMWH	ARR	RRR	NNT
PROM	0.03	0.037	−0.007	−0.244	N/A
OA	0.048	0.037	0.011	0.222	94
PTD	0.024	0.065	−0.041	−1.722	N/A
PL < 12 GW	0.095	0.13	−0.034	−0.361	N/A
GH	0.048	0.093	−0.045	−0.944	N/A
FGR	0.036	0.12	−0.085	−2.37	N/A
PA	0.012	0.009	0.003	0.222	378
HELLP	0.0	0.019	−0.019	0.0	N/A
SB	0.006	0.0	0.006	1.0	168
DVT	0.012	0.028	−0.016	−1.333	N/A

Abbreviations: PC—Pregnancy Complications; FVL—Factor V Leiden Mutation; FII—FII G20210A Mutation; PAI-1—Plasminogen Activator Inhibitor Mutation; MTHFR—Methylenetetrahydrofolate Reductase Mutation; CG- Control Group; PROM—Premature Rupture of Membranes; OA—Oligohydramnios; PTD—Preterm Delivery; FGR—Fetal Growth Restriction; PA—Placental Abruption; HELLP—Hemolysis, Elevated Liver Enzymes, Low Platelet Count; PE—Preeclampsia; SB—Stillbirth; GH—Gestational Hypertension; PL < 12 GW—Pregnancy Loss Before 12th Gestational Week; DVT—Deep Vein Thrombosis; ARR—Absolute Risk Reduction; RRR—Relative Risk Reduction; NNT—Number Needed To Treat. N/A—Not Applicable; value could not be calculated due to negative absolute risk reduction (ARR) or division by zero.

**Table 6 medicina-61-01131-t006:** Frequency of Pregnancy Complications in Factor V Leiden Mutation Carriers according to Low-Molecular-Weight Heparin Therapy Status.

Pregnancy Complication	Total N	LMWH (N = 57) %	No LMWH (N = 19), %	Fisher’s *p*
PROM	5	2 (3.5)	3 (15.8)	0.09
OA	4	2 (3.5)	2 (10.5)	0.25
PTD	2	1 (1.7)	1 (5.3)	0.43
FGR	9	2 (3.5)	7 (36.8)	0.0005
PA	0	0	0	1.0
HELLP	2	1 (1.7)	1 (5.3)	0.43
PE	3	0	3 (15.8)	0.013
SB	1	0	1 (5.3)	0.25
GH	10	4 (7.0)	6 (31.6)	0.012
RPL	10	4 (31.6)	6 (31.6)	0.012
DVT	3	1 (10.5)	2 (10.5)	0.15

Abbreviations: LMWH—Low-Molecular-Weight Heparin; PROM—Premature Rupture of Membranes; OA—Oligohydramnios; PTD—Preterm Delivery; PL—Pregnancy Loss; GDM—Gestational Diabetes Mellitus; GH—Gestational Hypertension; FGR—Fetal Growth Restriction; PA—Placental Abruption; HELLP—Hemolysis, Elevated Liver Enzymes, Low Platelet Count; SB—Stillbirth; DVT—Deep Vein Thrombosis.

**Table 7 medicina-61-01131-t007:** Multivariable Logistic regression analysis of Low-Molecular-Weight Heparin therapy and other predictors in relation to overall pregnancy complications.

Variable	OR (95% CI)	*p* Value
LMWH therapy (with vs. without)	1.071 (0.894–1.281)	0.462
Maternal age	1.036 (0.975–1.100)	0.250
BMI	1.008 (0.908–1.118)	0.885
IVF pregnancy	0.974 (0.505–1.881)	0.938
D-dimer (1st trimester)	1.070 (0.894–1.281)	0.462

## Data Availability

The raw data supporting the conclusions of this article will be made available by the authors on request.

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
