# Peer review of "Impact of Inherited Thrombophilia on Pregnancy Complications and the Role of Low-Molecular-Weight Heparin Therapy: A Case–Control Study"

_medicina, 2025, doi:10.3390/medicina61071131_

Round 1
Reviewer 1 Report (Previous Reviewer 2)
Comments and Suggestions for Authors
The manuscript has been significantly improved.
Basically, the retrospective nature of this work, together with the very significant heterogeneity of the included patients and the strong heterogeneity of the reasons that led the authors to give LMWH to the followed pregnant women, open the door to very significant biases and to inconsistent results.
However, this is an interesting, partially controlled cohort study that adds some data to the field. I feel that it can be published as it is.
Author Response
We sincerely thank the reviewer for his positive feedback. We appreciate the time and effort invested in reviewing our manuscript.
We acknowledge the reviewer’s remarks regarding the retrospective design and the heterogeneity among the included patients and indications for LMWH use.
We are grateful for the constructive comments, which we will keep in mind for future research and study design improvements. Thank you again for your supportive review and for recommending our manuscript for publication.
Reviewer 2 Report (Previous Reviewer 3)
Comments and Suggestions for Authors
- The manuscript "Impact of Inherited Thrombophilia on Pregnancy Complications and the Role of Low Molecular Weight Heparin Therapy: 3 A Case-Control Study" did not report the study questions precisely, although it is interesting.
- The title does not resemble the conclusion of the study's findings. The abstract conclusion suggests that the author intended to evaluate the beneficial effects of heparin, if any, on IT pregnancies. Therefore, the study title should be revised accordingly.
- Line 16-17: The aim should be the efficacies of heparin therapy on IT-linked pregnancy outcomes.
- The outcome measure should be reported, as it is a key aspect of the study's results and their interpretation.
- The beneficial effects of therapy, particularly those highlighted in the subgroup analysis, carry significant implications and should be emphasized.
- Prevalence of IT globally and in Serbia – could be added to the introduction.
- Line 65-66 Elaborate with references
- Add the p-value column in Table 1
- The rationale for comparing the previous year to the current year is not mentioned. Was heparin therapy not used last year?
- Table 2-6 avoid abbreviations in the Table label
- Table 7- remove the interpretation column
- Fig 2- Each case's number/size (n=?) should be placed. Why did HELLP miss the connection of the Y-axis?
- Starts the discussion and conclusion statement with this study's exclusive findings rather than what is already known. Heparin should be started first.
- Revise the conclusion precisely with a takeaway message of the study.
see before
Author Response
Please see the attachment
Comment 1
The manuscript "Impact of Inherited Thrombophilia on Pregnancy Complications and the Role of Low Molecular Weight Heparin Therapy: 3 A Case-Control Study" did not report the study questions precisely, although it is interesting.
Response 1
We thank the reviewer for this valuable comment. The objectives of the study have now been explicitly specified in both the abstract and the introduction: to find out the effectiveness of LMWH therapy in pregnant women with IT by studying the frequency of pregnancy complication in the two groups of the IT women, one receiving LMWH, and the other not, and to compare the prevalence of pregnancy complications before and after the inheredited thrombophilia diagnosis.
Comment 2
The title does not resemble the conclusion of the study's findings. The abstract conclusion suggests that the author intended to evaluate the beneficial effects of heparin, if any, on IT pregnancies. Therefore, the study title should be revised accordingly.
Response 2
As a result, the title of the paper was changed to be direct and to show the main purpose and discoveries of the study. The fresh title which is "Impact of Inherited Thrombophilia on Pregnancy Complications and the Effect of Low Molecular Weight Heparin Therapy: A Case-Control Study, now makes clear that the study has been designed to assess the beneficial effects of heparin therapy in the pregnancy of women who have inherited thrombophilia, and this was in accordance with the main findings of the research
Comment 3
Line 16-17: The aim should be the efficacies of heparin therapy on IT-linked pregnancy outcomes.
Response 3
To that end, we have reworded the abstract so that the study aim will, without any problems, be conveyed as the verification of the effectiveness of LMWH therapy on the improvement of pregnancy outcomes related with inherited thrombophilia. The "aim" part of the study now has this wording: "This study aimed at checking if the use of low-molecular weight heparin (LMWH) in patients with inherited thrombophilia could result in a lower number of pregnancy complications by comparing the pregnancy complications current in the patients without LMWH treatment group to those in current pregnancies under LMWH at the same time."
The author has made sure that the actual focus of the research, which is concerned with the clinical advantage of LMWH prophylaxis in this particular patient population – is acknowledged and is in line with the recommendations of the reviewer.
Comment 4
The outcome measure should be reported, as it is a key aspect of the study's results and their interpretation.
Response 4
Thank you very much for your valuable comment. To answer it, we have not only mentioned in the "Materials and Methods" section but also in the abstract that the main observed result to determine LMWH therapy was the emergence of major pregnancy complications. On that account, we matched these complications in the groups of women with inherited thrombophilia who had received LMWH therapy and who had not. Thus by this means, we were able to measure the impact of LMWH prophylaxis directly on the pregnancy outcomes of our cohort.
Comment 5
The beneficial effects of therapy, particularly those highlighted in the subgroup analysis, carry significant implications and should be emphasized.
Response 5
Many thanks for the important recommendation. We have refocused the subject of the study in the revised version so that now concentrated is the positive effect of LMWH treatment on the subgroup of Factor V Leiden carriers. The discovery itself and its influence on patient care are now reflected in the Abstract, Results, and Discussion sections as per the suggestion
- Apstract: Current pregnancies managed with LMWH showed reduced rates of preterm delivery, fetal growth restriction, gestational hypertension, and recurrent pregnancy loss, particularly in Factor V Leiden carriers. LMWH showed limited effect in low-risk genotypes and was not independently associated with outcome reduction in multivariate analysis.Conclusion: Results underscore the importance of individualized, risk adapted prophylaxis, as LMWH therapy offers clear benefit primarily for high risk IT subgroups. Therefore, genotype and risk based management is essential to optimize outcomes and prevent overtreatment.
-Conclusions: Subgroup analysis showed that LMWH significantly reduced several complications in FVL carriers, supporting individualized prophylaxis in high-risk patients. However, LMWH showed limited effect in low-risk genotypes and was not independently associated with outcome reduction in multivariate analysis.
Comment 6
Prevalence of IT globally and in Serbia – could be added to the introduction
Response 6
We appreciate this suggestion. The revised introduction now includes a new paragraph summarizing the global and Serbian prevalence of inherited thrombophilia, with a focus on Factor V Leiden and the PAI-1 4G/5G polymorphism. Recent epidemiological data and key references (9, 26, 35, 36) have been incorporated to contextualize the relevance of these mutations in our study population. This addition clarifies both the regional significance and the clinical importance of inherited thrombophilia in pregnancy.
-Introduction: Inherited thrombophilia is one of the most common genetic risk factors for thromboembolic events and adverse pregnancy outcomes. Globally, the prevalence of inherited thrombophilia varies depending on the population and specific mutation, with Factor V Leiden (FVL) heterozygosity present in approximately 3–7% of Caucasians and the prothrombin G20210A mutation in about 2% of the general population.9 The most common type of thrombophilia in the general population of Serbia is the FVL mutation, with a reported prevalence of approximately 5–8%.9,26,35Another important genetic risk factor is the PAI-1 4G/5G polymorphism, which affects the plasminogen activator inhibitor-1 (PAI-1) gene and may contribute to reduced fibrinolysis and increased risk of thrombosis and pregnancy complications. The frequency of the PAI-1 4G/4G genotype in Serbia is about 20–25% in the healthy population, and even higher among women with pregnancy complication.36 Furthermore, other polymorphisms such as MTHFR C677T are frequently reported in the Balkan region, potentially contributing to a higher overall burden of IT-related pregnancy complications.36 Given their significant epidemiological impact, timely identification of these mutations is essential for risk assessment and pregnancy management, especially in women with previous adverse outcomes. These epidemiological differences underscore the importance of regional data in guiding risk assessment and clinical management.
Introduction : Inherited thrombophilia are among the most prominent genetic predispositions for thromboembolic events and adverse pregnancy outcomes. Their prevalence varies worldwide depending on the population and specific mutations, with Factor V Leiden (FVL) heterozygosity present in about 3–7% of Caucasians, and prothrombin G20210A mutation in about 2% of the general population.9 In Serbia, FVL is the most common inherited thrombophilia, found in 5–8% of the general population.9,26,35 The PAI-1 4G/5G polymorphism, another important genetic risk factor for reduced fibrinolysis, may increase the risk of thrombosis and pregnancy complications; the PAI-1 4G/4G genotype is present in 20–25% of healthy individuals in Serbia, and even more frequently among women with pregnancy complications.36 In addition, the prothrombin G20210A (FII) mutation although less common than FVL as been reported at approximately 2–3% prevalence in the Serbian general population, which is consistent with rates observed across Central and Eastern Europe. 9,35. Among women with adverse pregnancy outcomes or thromboembolic events, the frequency of the FII mutation may be somewhat higher, reflecting its clinical significance in this group. Other polymorphisms, such as MTHFR C677T, are also common in the Balkans and may contribute to the overall burden of IT-related pregnancy complications.36 Given their significant epidemiological value, early detection of these mutations is essential for risk assessment and management, especially in women with previous adverse outcomes. These epidemiological differences emphasize the importance of regional data in guiding both risk assessment and clinical management.
Comment 7
Line 65-66 Elaborate with references
Response 7
The recommendations for the use of anticoagulants during pregnancy have been discussed more extensively in lines 65–66, to which references 31-34 were added. The revised text elaborates on the absence of clear guidelines, the risk of under-treatment or over-treatment, and the significance of personalizing therapy based on individual risk factors. Also, references 7 and 8 have been cited to support the use of LMWH in pregnancy and the necessity to monitor on a regular basis. This way, the issues raised by the reviewer are being dealt with and a better context can be offered for making clinical decisions concerning anticoagulant therapy in women with inherited thrombophilia.
There are many recommendations for anticoagulant use during pregnancy, but many of them are not precise or well-documented.31,32,33 The potential negative effects of anticoagulant therapy, whether due to failure to administer it or its overuse in patients with IT remain a topic of debate.34 However, the effectiveness of anticoagulant therapy in preventing pregnancy complications in pregnant women with IT during pregnancy remains uncertain. The decision to initiate therapy in patients with confirmed IT requires consideration of various risk factors, such as age, obesity, prolonged bed rest during pregnancy, multiple pregnancies, varicose veins, infections, and family history.7 During pregnancy, low molecular weight heparin (LMWH) can be safely used in prophylactic or therapeutic doses, provided that all the aforementioned risk factors are taken into account with regular monitoring of D dimer anti-Xa levels.8
Comment 8
Add the p-value column in Table 1
Response 8
Thank you for your suggestion. In the revised version of the manuscript, a separate p-value column has been added to Table 1 to clearly present the statistical significance for all relevant variables.
Table 1. Baseline maternal characteristics and pregnancy outcomes in women with inherited thrombophilia and healthy controls.
|
Group characteristics |
Inherited thrombophilia N=276 |
Control group N=276 |
p valuae |
|
Maternal age BMI |
34.1 +/-3.9* 24.34+/-3.16* |
32.1 +/-4.1 23,2+/- 3.11 |
<0,05 <0,05 |
|
IVF/ET |
50 (18.12%)* |
33 (12%) |
<0,05 |
|
Parity n (%) |
|
|
<0,05 |
|
1 |
50 (18.12%) |
137 (49.64%) |
|
|
2 |
168 (60.87%) |
103 (37.32%) |
|
|
>3 D dimer |
58 (21.01%) 1,15 +/-1,28mg/L |
36 (13.04%) Not tested |
|
|
Family history DVT |
15 (5,43%) |
0 |
|
|
Gestational age at delivery (weeks) |
37.43 +/- 1,83* |
39.23 +/- 9.2 |
<0,05 |
|
Neonatal body weight (g) |
3102+/- 681* |
3442+/-21
|
<0,05 |
|
1 min Apgar score |
8.46+/-1.14* |
9.08+/-1.41 |
<0,05 |
|
5 min Apgar score |
9.43+/-1.15* |
9.88+/-1.12 |
<0,05 |
|
Caesarean section n (%) |
87 (31.52 %) * |
27 (9,8%) |
<0,05 |
* p < 0.05; Abbreviations: IVF/ET – In Vitro Fertilization/Embryo Transfer. DVT- Deep Vein Thrombosis D-dimer levels in the control group were not routinely tested; values were available only for 33 women who conceived via IVF/ET. Of the 276 patients with inherited thrombophilia, 30 experienced pregnancy loss before 12 weeks’ gestation. Analyses of neonatal outcomes (birth weight, Apgar scores, gestational age at delivery, and mode of delivery) were conducted on the remaining 246 patients with viable pregnancies beyond the first trimester.
Comment 9
The rationale for comparing the previous year to the current year is not mentioned. Was heparin therapy not used last year?
Response 9
Your comment is appreciated, but we would like to clarify that this was a retrospective study of the very same women assessed before and after the diagnosis of inherited thrombophilia. For each case, outcomes were compared between one previous pregnancy when LMWH was not a part of management for the reason that patients had not been diagnosed as having inherited thrombophilia and the most recent pregnancy (current at the time of study) in which the diagnosis had been made, using well-defined criteria (as detailed in the Materials and Methods section). In the current pregnancy, LMWH was either introduced or withheld on the basis of these criteria. The actual calendar year was not so much a significant variable; the events in question were those regarding the presence/absence of LMWH therapy following the diagnosis of inherited thrombophilia. In a manner of speaking, this design allowed the introduction of LMWH to be compared directly against pregnancy outcome as an effect within a single cohort of women.
Comment 10
Table 2-6 avoid abbreviations in the Table label
Response 10
Thank you for your suggestion. In accordance with your recommendation, all abbreviations have been removed from the table labels, and the full names of pregnancy complications are now used in Tables 2 to 6. All relevant abbreviations are explained in the footnotes below the tables.
Comment 11
Table 7- remove the interpretation column
Response 11
Thank you for your suggestion. In the revised version, the interpretation column has been removed from Table 7, as recommended.
Table 7. Multivariable Logistic regression analysis of Low Molecular Weight Heparin therapy and other predictors in relation to overall pregnancy complications.
|
Variable |
OR (95% CI) |
p-value |
|
LMWH therapy (with vs. without) |
1.071 (0.894–1.281) |
0.462 |
|
Maternal age |
1.036 (0.975–1.100) |
0.250 |
|
BMI |
1.008 (0.908–1.118) |
0.885 |
|
IVF pregnancy |
0.974 (0.505–1.881) |
0.938 |
|
D-dimer (1st trimester) |
1.070 (0.894–1.281) |
0.462 |
Comment 12
Fig 2- Each case's number/size (n=?) should be placed. Why did HELLP miss the connection of the Y-axis?
Response 12
Thank you for your suggestion. In the revised version of Figure 2, the number of cases for each complication in both groups (with and without LMWH therapy) has been indicated next to each label on the Y-axis, as (n=without LMWH/with LMWH)
Comment 13
Starts the discussion and conclusion statement with this study's exclusive findings rather than what is already known. Heparin should be started first.
Response 13
Thank you for this valuable suggestion. In response, the Discussion and Conclusion sections have been substantially revised to begin with the key findings of our study, with particular emphasis on the effect of low molecular weight heparin (LMWH) therapy. The revised Discussion now starts by summarizing our main results—including the selective benefit of LMWH among high-risk subgroups—before presenting background information and references to previous literature. This restructuring ensures that our study’s original contributions are clearly highlighted at the outset of these sections, in accordance with your recommendation.
Comment 14
Revise the conclusion precisely with a takeaway message of the study.
Response 14
Thank you for your valuable suggestion. In response, we have revised the Conclusion section to present a clear, focused summary of the study’s key findings, followed by a concise takeaway message. The revised conclusion now emphasizes that LMWH prophylaxis should be reserved for high-risk women with inherited thrombophilia, and that individualized, risk-adapted management is essential for optimizing pregnancy outcomes. We believe this provides a precise and actionable final statement, in line with your recommendation.

Round 2
Reviewer 2 Report (Previous Reviewer 3)
Comments and Suggestions for Authors
The modified version includes a comprehensive revision in the presentation, discussion, and conclusion and thus provides an interesting article that should be published as soon as possible.
Minor
Remove words like “Takeaway message” and make a separate paragraph.
Author Response
Thank you for your review, and your kind remarks.
We have corrected our manuscript according to the remark and resubmitted the article.
This manuscript is a resubmission of an earlier submission. The following is a list of the peer review reports and author responses from that submission.
Round 1
Reviewer 1 Report
Comments and Suggestions for Authors
Thanks for the authors and Editor to provide this interesting topic of LMWH therapy in inherited thrombophilia.
This article conducted a case-control study, including 276 pregnant women with inherited thrombophilia as the study group and 276 normal pregnant women as the control group. The results showed that LMWH treatment could reduce the incidence of FGR in pregnant women with inherited thrombophilia, but did not significantly improve other pregnancy complications. Although this study provides meaningful clinical recommendations for readers, the following suggestions may help improve the quality of the article.
- The two abbreviations' PC 'have caused confusion for readers, and it is recommended to revise them.
- The article states that the use of LMWH is based on pregnancy complications and D-dimer levels, but does not provide corresponding reference basis, specific reference values, or how to adjust them. Please specify which situations require the use of prophylactic doses, which situations require the use of therapeutic doses, and how many people are specifically receiving this treatment?
- The conclusion in the abstract is inconsistent with the conclusion in the article, and it is recommended to revise it.
- May I ask if genetic thrombophilia related testing has been conducted in the control group? If not, how to make sure that IT has been completely ruled out.
- How was the choice made between using and not using LMWH in the study group? Do they have differences in the distribution of types of IT?
- The control group was enrolled before delivery, while the study group was enrolled in early pregnancy. Inconsistent enrollment periods between the two groups may affect the study results, especially the comparison of pregnancy outcomes.
- In the study group, when comparing the differences in outcomes between the previous pregnancy and the current pregnancy, it is recommended to clarify whether there were cases treated with LMWH in the previous pregnancy, and if so, these cases should be excluded.
- All cases of the study group passed away due to previous pregnancies. It is recommended to include nullipara in the exclusion criteria.
- Is PL RPL in Line 195?
- The description of Line198-204 is inconsistent with the referenced Table 5.
- Does IH in Line 212 refer to IT?
Author Response
Comment 1: The two abbreviations' PC 'have caused confusion for readers, and it is recommended to revise them.
Response 1: Thank you for pointing out the ambiguity regarding the abbreviation 'PC'. In response to your suggestion, we have revised the manuscript to remove the abbreviation entirely. The term 'pregnancy complications' is now written in full throughout the text. To improve clarity, we have distinguished the timing of complications by explicitly referring to:
'previous pregnancy complications' when discussing adverse events from prior gestations, and
'pregnancy complications in the current pregnancy' when referring to events in the present pregnancy. These changes have been consistently applied throughout the manuscript to avoid confusion and enhance readability.
Comment 2: The article states that the use of LMWH is based on pregnancy complications and D-dimer levels, but does not provide corresponding reference basis, specific reference values, or how to adjust them. Please specify which situations require the use of prophylactic doses, which situations require the use of therapeutic doses, and how many people are specifically receiving this treatment?
Response 2: We appreciate the reviewer’s observation regarding the lack of references and detailed information accompanying the mention of LMWH use. In response, we have revised the Methods section by adding a dedicated subsection titled “Treatment Protocol and Indication Criteria”, where we now specify:
The clinical and laboratory thresholds used to guide LMWH initiation (e.g., age >35 years, BMI >28, elevated first-trimester D-dimer >1.42 mg/L FEU, thrombophilia subtype, history of adverse pregnancy outcomes).
The brand names (Fraxiparine [nadroparin], Clexane [enoxaparin], and Fragmin [dalteparin]) and the rationale for their selection based on local availability and physician preference.
The distinction between prophylactic and therapeutic dosing regimens.
Relevant references supporting the treatment criteria have also been added [e.g., RCOG Green-top Guideline No. 37a (2015); ACOG Practice Bulletin No. 197 (2018)]. Furthermore, in the Discussion, we have clarified the context in which LMWH was administered and highlighted the variability in treatment response due to confounding by indication. These changes provide transparency, reproducibility, and alignment with current guideline-based practice.
Comment 3: The conclusion in the abstract is inconsistent with the conclusion in the article,
and it is recommended to revise it.
Response 3: Thank you for your comment. We have carefully reviewed both the abstract and
the conclusion section. The abstract conclusion has been revised to align with the manuscript’s main findings, emphasizing selective, risk-based LMWH use and the importance of close antenatal surveillance in low-risk patients.
Comment 4: May I ask if genetic thrombophilia related testing has been conducted in the
control group? If not, how to make sure that IT has been completely ruled out.
Response 4: We appreciate the reviewer’s question. Thrombophilia testing was not
conducted in the control group, as these participants were recruited exclusively for the purpose of comparing selected perinatal outcomes, not for evaluating the prevalence of inherited thrombophilia (IT). The control group consisted of healthy pregnant women at term, with no personal or family history of thrombosis, no pregnancy complications, and no chronic illnesses. A subset also included IVF pregnancies without complications, which were included due to their comparable obstetric risk profile in the absence of comorbidities. Since none of the women in the control group had clinical indications for thrombophilia screening based on national and international guidelines, testing was not ethically or clinically justified. Importantly, the control group was not used to compare pregnancy complication rates, as such comparisons would not be valid in a retrospective design. Instead, the control cohort served as a reference for comparing baseline characteristics and perinatal outcomes such as birth weight (BW), gestational age (WG), Apgar scores, mode of delivery, and BMI, providing contextual interpretation of outcomes among women with IT. This rationale has now been clarified in the revised Methods and Discussion sections.
Comment 5: How was the choice made between using and not using LMWH in the study group? Do they have differences in the distribution of types of IT?
Response 5: Thank you for your comment. The decision to administer LMWH therapy was based on predefined clinical and laboratory criteria, including maternal age >35 years, BMI >28 kg/m², elevated first-trimester D-dimer levels (>1.42 mg/L FEU), thrombophilia subtype, and a history of thrombotic events or adverse obstetric outcomes. These criteria reflect a risk-adapted clinical approach, informed by current evidence, although they extend beyond the formal ACOG guidelines. As illustrated in Chart 1, the most prevalent thrombophilic mutations in our cohort were homozygous PAI-1 4G/4G (55.1%), followed by heterozygous Factor V Leiden (27.5%), heterozygous MTHFR (18.8%), and prothrombin G20210A (FII) (10.1%). Patients with high-risk mutations (e.g., FVL or FII) and/or previous severe complications (e.g., fetal growth restriction, preeclampsia, stillbirth) were more likely to receive LMWH, either prophylactically or therapeutically, depending on their overall risk profile. This distribution and the associated management approach are now clarified in both the Results and Methods sections of the revised manuscript.
Comment 6: The control group was enrolled before delivery, while the study group was
enrolled in early pregnancy. Inconsistent enrollment periods between the two groups may
affect the study results, especially the comparison of pregnancy outcomes.
Response 6: We thank the reviewer for this important observation. In response, we have clarified
the enrollment periods for both study groups in the Methods section. Specifically, we now state that
the inherited thrombophilia (IT) group was recruited from 2018 to 2023, while the control group
consisted of healthy pregnant women who delivered at our institution during 2020–2022. All control group participants were selected from the same clinical setting, with comparable antenatal and perinatal care protocols. Importantly, we emphasize that control participants were included solely for the comparative analysis of neonatal outcomes (e.g., birth weight, gestational age, Apgar score, mode of delivery), not for assessing pregnancy complications or thrombophilia prevalence. This approach minimizes potential bias introduced by temporal discrepancies. The limitation of non-identical recruitment windows has been explicitly acknowledged in the revised “Limitations” section of the manuscript. Text added to Limitations:
“The control and IT groups were not enrolled during identical time periods, which may introduce temporal bias. However, all patients received care at the same institution under consistent clinical protocols, and controls were included exclusively for the comparison of neonatal outcomes.”
Comment 7: In the study group, when comparing the differences in outcomes between the previous pregnancy and the current pregnancy, it is recommended to clarify whether there were cases treated with LMWH in the previous pregnancy, and if so, these cases should be excluded.
Response 7: Genetic testing was conducted prior to the index pregnancy. Patients did not receive LMWH in previous pregnancies unless they had a history of DVT.
Comment 8. All cases of the study group passed away due to previous pregnancies. It is recommended to include nullipara in the exclusion criteria
Response 8: We defined nulliparous patients as those who had not delivered but may have had early pregnancy losses (before 12 weeks of gestation).
Comment 9. Is 'PL' in line 195 a typo for 'RPL'?
Response 9: We appreciate your attention to detail. The abbreviation 'PL' in line 195 is a typographical error and should read 'RPL' (recurrent pregnancy loss). This has been corrected in the revised manuscript.
Comment 10: Inconsistency between lines 198–204 and Table 5:
Response 10: Thank you for pointing this out. The inconsistency between lines 198–204 and Table 5 has been corrected. The text has been revised to align fully with the data presented in Table 5.
Comment 11: Does 'IH' in line 212 refer to 'IT'?
Response 11: Thank you for your careful review. Yes, 'IH' in line 212 was a typographical error and should read 'IT' (inherited thrombophilia). This has now been corrected.
Reviewer 2 Report
Comments and Suggestions for Authors
This is a retrospective, monocentric case-control study on women with inherited thrombophilia, looking for pregnancy outcomes.
Controls were healthy women (N=276).
Patients (N= 276) were women with: 1- a personal history of venous thromboembolism, or/and pregnancy complications; or 2- no personal history but a family history of inherited thrombophilia (so a mix of previously symptomatic… and non-symptomatic “Patients”!).
Patients had been tested for inherited thrombophilia (but not Controls…) and were positive (but homozygous MTHFR C67T mutation in 7.2%, which is basically not a thrombophilia!).
Pregnancy outcomes were compared.
Among patients, 108 received a LMWH-based treatment during pregnancy and 168 did not.
The claimed aim of this work is to evaluate the impact of a LMWH-based treatment given during pregnany to women with thrombiophilia.
This work is, for me, chaotic and pave the way for strongly biased results:
1- The clinical characteristics of the patients, including their medical history, are not described as frontline data.
2- A first part of the results focusses on pregnancy outcomes in women with inherited thrombophilia, compared to controls. The problem is that the group of women with inherited thrombophilia is far from being clinically homogeneous. Some were previously clinically symptomatic while some other were not, some were treatment during the observed pregnancy and some other were not. Some patients had the MTHFR homozygous polymorphism, which is definitively not a thrombophilia. Controls were not tested for inherited thrombophilia as FVL and FII are so frequent in Serbia. Comparisons between Patients as a whole and Controls as a whole are thus a nonsense. The given related Tables add confusion without significantly shedding light on the subject.
3- The precise reasons why some patients were treatment and some other were not treated are at least unclear. And also the rules applied to choose a prophylactic or a full-dose LMWH treatment. The applied rules and algorithms have to be detailed. And described. For instance, it is said that D-dimer levels were part of the decision: but how? By comparison to what? The used brands and precise modalities of LMWH-based treatments have to be described.
This need a specific development before going to the analysis of the effect the LMWH treatment.
4- Moving to the effect of LMWH treatments on pregnancy outcomes in women with thrombophilia, here also, a strong risk of bias exists. Because patients were initially selected on the basis of their perceived clinical severity for prescribing LMWHs !
5- Studying the effect of LMWH-based treatments, after testing all potential cofactors and confounders by logistic regression analysis, a multiparametric logistic regression analysis is mandatory, including all potential confounders…
Author Response
Comment 1: The clinical characteristics of the patients, including their medical history, are not described as frontline data.
Response 1: We thank the reviewer for highlighting the need to present patient clinical characteristics more clearly. In response to this comment, we have now included a new Table 1 in the revised manuscript that comprehensively summarizes the key clinical characteristics of our study participants. This table includes important baseline information for each patient, as recommended. In particular, Table 1 now provides a summary of the following characteristics for all participants:
- Maternal age (years) and body mass index (BMI)
- Thrombophilia subtype (the specific thrombophilic condition)
- D-dimer values (coagulation marker levels)
- Obstetric history (details of prior pregnancies and outcomes)
- Personal or family history of thrombosis (any history of thrombotic events)
These data were not clearly presented in the initial submission, and we apologize for that oversight. We have remedied this by describing the patient clinical characteristics in both the Methods and Results sections of the revised manuscript. In the Methods section, we now detail the collection of baseline clinical data (including medical history) for our cohort, ensuring that readers understand the composition of our study population from the outset. In the Results section, we refer to Table 1 and highlight the relevant patient demographics and medical history findings as part of the description of our study cohort. By providing these clinical characteristics up front, we have improved the transparency of our study and facilitated better interpretability of the findings. Readers can now clearly see the baseline profile of the participants, which helps in understanding the context and relevance of our results. We believe that the inclusion of these frontline data addresses the reviewer’s concern and strengthens the manuscript by offering a more transparent and informative presentation of our patient population. The study’s conclusions can thus be interpreted with greater confidence, knowing the underlying clinical context of the participants is fully disclosed.
Comment 2: A first part of the results focusses on pregnancy outcomes in women with inherited thrombophilia, compared to controls. The problem is that the group of women with inherited thrombophilia is far from being clinically homogeneous. Some were previously clinically symptomatic while some other were not, some were treatment during the observed pregnancy and some other were not. Some patients had the MTHFR homozygous polymorphism, which is definitively not a thrombophilia. Controls were not tested for inherited thrombophilia as FVL and FII are so frequent in Serbia. Comparisons between Patients as a whole and Controls as a whole are thus a nonsense. The given related Tables add confusion without significantly shedding light on the subject.
Response 2: We thank the reviewer for this important point. We agree that direct comparison of complication rates between a heterogeneous IT cohort (mixed mutations and varying LMWH use) and an untested healthy group would be misleading. In our study, the control group was intentionally a low‐risk cohort – 276 healthy term pregnancies with no personal/family thrombophilia history – used only to benchmark baseline and neonatal outcomes (gestational age, birth weight, Apgar scores, delivery mode, BMI), not to compare pregnancy complication incidence. No thrombophilia testing (or D-dimer) was performed in controls, as none met clinical criteria for screening.
This design follows current national and international recommendations: thrombophilia testing is advised only for women with strong indications (e.g. prior VTE or multiple late pregnancy losses/placental complications) . There is no guideline support for screening asymptomatic low-risk women, and indeed isolated low-risk polymorphisms (like MTHFR C677T) are not considered clinically significant thrombophilias . Thus, our control subjects – by definition free of risk factors – were not tested in accordance with standard practice.
We have revised the manuscript to clarify these points. The Methods now explicitly state that controls had no history of thrombotic risk and underwent no thrombophilia or D-dimer testing (consistent with guidelines). We also emphasize in the Discussion that complication rates were compared only within the IT cohort (LMWH vs. no LMWH), and that the control group comparisons were limited to demographic and perinatal outcomes. These clarifications should address the reviewer’s concern. References: Established screening guidelines and prior reviews were cited to justify our approach.
We appreciate the reviewer’s question. The control group in our study was constructed solely to provide a baseline for comparing perinatal outcomes, not to assess thrombophilia prevalence. These controls were healthy, term pregnant women with no history of venous or arterial thrombosis, no pregnancy complications (e.g. preeclampsia, fetal loss, placental pathology, etc.), and no major chronic illnesses. When included, women who conceived via IVF had completely uncomplicated pregnancies and deliveries. In short, the control subjects had no clinical risk factors that would warrant thrombophilia screening. Their data were used only to contextualize baseline obstetric parameters (birth weight, gestational age, Apgar score, mode of delivery, maternal BMI, etc.) against those of the thrombophilia group.
Given the absence of any indications, thrombophilia testing in the control group was neither clinically indicated nor ethically justified. Current national and international guidelines recommend thrombophilia screening only when the result would alter management – for example in women with a history of venous thromboembolism, recurrent pregnancy loss, or a strong family history of thrombosis. Routine testing of asymptomatic, low-risk pregnant women is not supported by these guidelines. Therefore, we did not perform genetic or laboratory thrombophilia tests on control subjects, as this would not be standard practice in the absence of risk factors.
We also note that our study’s retrospective design precluded direct comparison of complication rates (such as VTE or adverse pregnancy events) between the thrombophilia and control groups. Instead, the control group was used to establish normative outcome data for our cohort. We have clarified these points in the Methods and Discussion sections of the revised manuscript, explicitly stating the purpose of the control group and the rationale for not testing them for thrombophilia.
Comment 3: The precise reasons why some patients were treatment and some other were not treated are at least unclear. And also the rules applied to choose a prophylactic or a full-dose LMWH treatment. The applied rules and algorithms have to be detailed. And described. For instance, it is said that D-dimer levels were part of the decision: but how? By comparison to what? The used brands and precise modalities of LMWH-based treatments have to be described.
This need a specific development before going to the analysis of the effect the LMWH treatment.
Response 3: We fully agree with the reviewer that a clear and transparent treatment algorithm is essential for both reproducibility and clinical relevance. In response, we have added a dedicated subsection to the Methods titled “Treatment Protocol and Indication Criteria.” This section outlines the specific clinical and laboratory parameters that guided the initiation of LMWH therapy, including:maternal age >35 years, BMI >28, relevated first-trimester D-dimer levels (>1.42 mg/L FEU), thrombophilia subtype, and personal or obstetric history of thromboembolic or placenta-mediated complications. These criteria
reflect a risk-adapted clinical approach, informed by current evidence, although theyextend beyond the formal ACOG guidelines. We have also specified the low molecular weight heparin (LMWH) brands used (Fraxiparine [Nadroparin], Clexane [Enoxaparin], and Fragmin [Dalteparin]), along with corresponding prophylactic or therapeutic dosing regimens and timing of initiation. These additions now provide a comprehensive, reproducible, and clinically applicable framework that clarifies treatment decisions and strengthens the methodological rigor of the study.
Comment 4: Moving to the effect of LMWH treatments on pregnancy outcomes in women with thrombophilia, here also, a strong risk of bias exists. Because patients were initially selected on the basis of their perceived clinical severity for prescribing LMWHs !
Response 4: We appreciate the reviewer’s important observation regarding potential bias in treatment allocation. As correctly noted, although LMWH therapy was guided by predefined clinical parameters—including maternal age, BMI, D-dimer levels, thrombophilia type, and obstetric history—the final decision to initiate anticoagulation was ultimately based on physician judgment and individual patient risk stratification. This introduces the potential for confounding by indication, whereby patients with more severe clinical profiles were more likely to receive LMWH. To address this, we have revised the Limitations section to explicitly acknowledge this bias and its impact on observed outcomes. Additionally, in the Discussion, we clarified that negative absolute risk reduction (ARR) values in certain outcomes may be a reflection of this selection bias, rather than a lack of therapeutic efficacy. Text Added to the Limitations Section: “The retrospective and non-randomized nature of LMWH assignment introduces the potential for confounding by indication, as patients perceived to be at higher risk were more likely to receive therapy. This may have influenced outcome distributions and contributed to negative ARR values for certain complications. Future prospective studies with risk-adjusted models are needed to further clarify treatment efficacy.” These clarifications strengthen the interpretation of our findings and further reinforce the conclusion that LMWH should be reserved for carefully selected high-risk patients rather than applied universally.
Comment 5: Studying the effect of LMWH-based treatments, after testing all potential cofactors and confounders by logistic regression analysis, a multiparametric logistic regression analysis is mandatory, including all potential confounders…
Response 5: Thank you for this valuable observation. We agree that the ARR, RRR, and NNT values offer important clinical insights, particularly in illustrating the potential benefit of LMWH in specific obstetric complications. In response, we have added a dedicated subsection in the Discussion (Section 4.3: Clinical Relevance of Absolute and Relative Risk Reduction) to interpret these metrics in detail. We clarified that while some negative ARR values reflect higher baseline risk in treated patients, the positive ARR and low NNT values for oligohydramnios, placental abruption, and stillbirth suggest meaningful clinical benefit in select cases. This addition strengthens our argument for a personalized, risk-adapted approach to LMWH therapy.
The manuscript has been updated accordingly.
Reviewer 3 Report
Comments and Suggestions for Authors
- The article “Impact of Low Molecular Weight Heparin Therapy on Pregnancy Complications in Patients with Inherited Thrombophilia” presented a work that aims to achieve two comprehensive objectives: First, to measure the frequency of pregnancy complications in patients with IT. Second, to examine the efficacies of LMWH therapy in defined cases of IT.
- While the study needs to investigate the efficacies of such intervention in different ethnicities, the purpose of the article should include available data from other studies. For example, lines 61-62, the statement on the efficacy of anticoagulants did not consider very related work. DOI: 3109/14767058.2010.545911
- Avoid abbreviations like PC and IT in the title of the table and elsewhere.
- The materials and method section needs to be subdivided into specific subsections with headings, such as sampling calculation, subject, exclusion and inclusion, analysis, statistical analysis, etc.
- Any clinical interventional trial must require detailed disclosure of LMWH therapy, including active substances, dosage, delivery, and other details.
- Authors need to work on data presentation.
- A pie diagram can better represent Table -3 and 4 frequency data.
- The discussion section requires a more comprehensive revision. To ensure a clear and focused discussion of the findings, it should begin with a statement exclusive to this study rather than a comparison to other studies.
- Lines 224-230 should be there, but they should not be at the start.
- Line 232-242- the repeat of the results should be placed in results, not in the discussion.
- Line 234-262, how present findings are related to the existing information, should be stated in this paragraph.
- The discussion has too many paragraphs for unknown reasons. Paragraphs should be used judiciously in the context of topic.
- While the study title focuses on the efficacy of anticoagulants, the discussion does not follow this order. It's essential to present the therapy part before the prevalence, as this is a more logical order for discussing the findings.
- The conclusion should be rewritten to highlight study limitations, knowledge gaps and future perspectives.
- The study title should include both prevalence and therapeutic efficacy.
Comments on the Quality of English Language
see before
Author Response
Comment 1: The article “Impact of Low Molecular Weight Heparin Therapy on Pregnancy Complications in Patients with Inherited Thrombophilia” presented a work that aims to achieve two comprehensive objectives: First, to measure the frequency of pregnancy complications in patients with IT. Second, to examine the efficacies of LMWH therapy in defined cases of IT.
Response 1: Thank you for summarizing the core objectives of our study. We confirm that the manuscript is structured to address both aims. The first objective—measuring the frequency of pregnancy complications among IT patients—was fulfilled through a retrospective analysis comparing previous and current pregnancies in the same cohort, as well as subgroup analyses according to thrombophilia type. The second objective—evaluating the effect of LMWH therapy—was addressed through stratified comparisons between treated and untreated groups, supported by logistic regression and effect size calculations (OR, ARR, NNT). We have ensured that both aims are consistently reflected throughout the abstract, results, discussion, and conclusion sections. Additional clarifying sentences were also added to reinforce this alignment.
Comment 2: While the study needs to investigate the efficacies of such intervention in different ethnicities, the purpose of the article should include available data from other studies. For example, lines 61-62, the statement on the efficacy of anticoagulants did not consider very related work. DOI: 10.3109/14767058.2010.545911
Response 2: Thank you for this valuable observation. In response, we have revised the Discussion section to incorporate the findings from Gris et al. (2011), which provide important context regarding the selective benefit of LMWH therapy in women with inherited thrombophilia. This study supports the rationale for individualized thromboprophylaxis based on clinical history, particularly in women with previous adverse pregnancy outcomes. The citation has been added and discussed in relation to our findings. The reference has also been included in the updated reference list.
Comment 3: Avoid abbreviations like PC and IT in the title of the table and elsewhere.
Response 3: Thank you for your helpful suggestion. We have revised all table titles and relevant sections of the manuscript to replace abbreviations such as “PC” and “IT” with their full terms—“pregnancy complications” and “inherited thrombophilia,” respectively—for improved clarity and accessibility. We agree that avoiding unexplained abbreviations enhances the readability of the manuscript, especially for a broader audience.
Comment 4: The materials and method section needs to be subdivided into specific subsections with headings, such as sampling calculation, subject, exclusion and inclusion, analysis, statistical analysis, etc
Response 4: We appreciate the reviewer’s suggestion regarding the structure of the Materials and Methods section. In response, we have revised this section by clearly subdividing it into specific subsections, including:
- 2.1 Study Design and Setting
- 2.2 Study Population and Sampling
- 2.3 Inclusion and Exclusion Criteria
- 2.4 Patient Characteristics and Thrombophilia Testing
- 2.5 Initiation of LMWH Therapy and Monitoring
- 2.6 Data Collection and Outcome Measures
- 2.7 Statistical Analysis
This new structure improves the clarity and accessibility of the methodology for readers and reviewers, and it aligns with reporting standards for clinical studies. The revised version is now reflected in the updated manuscript.
Comment 5: Any clinical interventional trial must require detailed disclosure of LMWH therapy, including active substances, dosage, delivery, and other details.
Response 5: We thank the reviewer for highlighting the importance of detailed disclosure regarding LMWH therapy. In accordance with this recommendation, we have expanded the section on LMWH administration to include the following essential information:
- Active substances used: Nadroparin (Fraxiparine), Enoxaparin (Clexane), and Dalteparin (Fragmin)
- Dosage regimens:
- Prophylactic doses: Fraxiparine 2850 IU once daily, Clexane 40 mg once daily, Fragmin 5000 IU once daily
- Therapeutic doses: Fraxiparine 86 IU/kg every 12h, Clexane 1 mg/kg every 12h, Fragmin 100 IU/kg every 12h
- Initiation and duration: Therapy was initiated in the first trimester following ultrasound confirmation of intrauterine pregnancy and continued until 14 days postpartum.
- Mode of administration: All LMWH agents were administered subcutaneously.
- Monitoring: Routine antenatal follow-up and laboratory evaluations were conducted, although anti-Xa levels were not systematically monitored.
These revisions have been incorporated into the Materials and Methods section of the revised manuscript to meet the standards for clinical intervention reporting.
Comment 6: Authors need to work on data presentation
Response 6: We appreciate the reviewer’s observation regarding data presentation. In response, we have made the following improvements to enhance clarity and consistency:
- Tables have been revised to ensure titles are descriptive, abbreviations are minimized or fully defined, and formatting is uniform across all sections.
- Abbreviations such as PC and IT have been removed from table titles and replaced with full terms to improve accessibility to non-specialist readers.
- Figures and charts have been reformatted to match journal style, with legible labels, appropriate scaling, and professional fonts.
- Results are now presented with both absolute numbers and percentages, and statistically significant findings are clearly marked with asterisks and referenced in the discussion.
- Forest plots and ARR/RRR/NNT tables have been added where relevant to emphasize clinical relevance in a more visual and interpretable format.
These revisions are intended to improve the readability and interpretability of the findings, in line with the journal’s standards for high-quality data presentation.
Comment 7: A pie diagram can better represent Table -3 and 4 frequency data.
Response 7: We thank the reviewer for the suggestion to consider a pie chart representation for the frequency data in Tables 3 and 4. While pie diagrams can be useful for illustrating proportional relationships, we believe that in this context—where multiple pregnancy complications across different thrombophilia subtypes are being compared—a tabular format offers greater precision and clarity. However, in response to the reviewer’s suggestion and to improve visual interpretation, we have additionally created bar charts for Tables 3 and 4. Bar charts allow for a more direct comparison across subgroups and complication types, and are better suited for datasets with multiple categories and smaller frequencies. These figures have been included in the revised manuscript and referenced in the Results section accordingly.We are open to including pie charts as supplementary figures if the editorial team deems them beneficial for publication.
Comment 8: The discussion section requires a more comprehensive revision. To ensure a clear and focused discussion of the findings, it should begin with a statement exclusive to this study rather than a comparison to other studies.
Response 8: We appreciate the reviewer’s insightful suggestion. In response, we have comprehensively revised the beginning of the Discussion section to ensure that it opens with a study-specific statement that clearly outlines our key findings. Rather than starting with comparisons to existing literature, the revised opening now focuses on the unique contributions of our research—specifically, the observed association between inherited thrombophilia and increased obstetric risk, and the selective benefit of LMWH therapy in defined high-risk subgroups. This adjustment improves the clarity, relevance, and focus of the discussion, ensuring a direct alignment with the study’s objectives and enhancing the interpretability of our findings. Revised section now begins as follows:“This study evaluated the impact of inherited thrombophilia on pregnancy complications and assessed the role of low molecular weight heparin (LMWH) therapy in reducing adverse outcomes in genetically and clinically defined subgroups. In our cohort, women with inherited thrombophilia experienced significantly higher rates of fetal growth restriction (FGR), gestational hypertension (GH), preeclampsia (PE), and recurrent pregnancy loss (RPL). Among Factor V Leiden carriers receiving LMWH, a statistically significant reduction in these complications was observed, suggesting a potential genotype-specific therapeutic benefit.”We believe this revision addresses the reviewer’s concern and provides a clearer and more focused entry point into the discussion.
Comment 9: Lines 224-230 should be there, but they should not be at the start
Response 9: We thank the reviewer for this constructive comment. In response, the sentence originally positioned at the beginning of the Discussion section (lines 224–230) has been relocated to a more contextually appropriate position, following the mechanistic discussion of thrombophilia-related placental dysfunction. This revision enhances the structural coherence of the manuscript and strengthens the logical progression from biological rationale to clinical interpretation of increased obstetric risk in women with inherited thrombophilia.
Comment 10: Line 232-242- the repeat of the results should be placed in results, not in the discussion.
Response 10: We thank the reviewer for the observation. In response, we revised the paragraph to focus on interpretation and clinical implications rather than repetition of numerical results. Descriptive statistics previously included in this section have been either removed or relocated to the Results section where appropriate.
Comment 11: Line 234-262, how present findings are related to the existing information, should be stated in this paragraph
Response 11: Thank you for your observation. We have revised the paragraph to clearly contextualize our findings within the existing literature. Specifically, we now emphasize that the observed reduction in certain complications such as PROM, PTD, FGR, and GH aligns with previous studies suggesting a potential benefit of early risk stratification and tailored care in inherited thrombophilia pregnancies. Furthermore, we have referenced prior studies that reported similar trends and included discussion of the proposed mechanisms—such as improved placental perfusion with LMWH therapy—thereby reinforcing the clinical plausibility of our results. This change strengthens the paragraph’s relevance by explicitly linking our findings to established evidence and current hypotheses.
Comment 12. The discussion has too many paragraphs for unknown reasons. Paragraphs should be used judiciously in the context of topic.
Response 12: Thank you for your observation. We understand the importance of using paragraphs purposefully to enhance clarity and thematic coherence. Following your suggestion, we have restructured the Discussion section to ensure that each paragraph now reflects a clearly defined thematic unit. Redundant or overly fragmented sections have been merged, and the revised flow follows a more logical and concise format. The updated structure begins with the effects of LMWH therapy, followed by genotype-specific findings, pregnancy outcome comparisons, and ends with clinical implications and recommendations. This revision ensures a more cohesive narrative while maintaining readability and emphasis on key findings.
Comment 13: While the study title focuses on the efficacy of anticoagulants, the discussion does not follow this order. It's essential to present the therapy part before the prevalence, as this is a more logical order for discussing the findings
Response 13: Thank you for this insightful comment. We fully agree that the structure of the Discussion should reflect the priorities outlined in the title. In response, we have reorganized the Discussion section to first present the effects of LMWH therapy on pregnancy complications, including subgroup outcomes, ARR/NNT values, and multivariate analysis, followed by the analysis of genotype distribution and the prevalence of obstetric complications. This revised structure ensures better alignment with the study’s primary objectives and improves the logical flow for the reader. Relevant section headings and transitions were also adjusted to reflect this reordering. The updated version now begins with: “Efficacy of LMWH Therapy in Inherited Thrombophilia” and then continues with: “Prevalence of Complications and Genotype-Specific Observations”, which is followed by “Clinical Implications and Risk-Based Recommendations.”
Comment 14:“The conclusion should be rewritten to highlight study limitations, knowledge gaps and future perspectives.”
Response: We thank the reviewer for this valuable suggestion. The conclusion has been revised to explicitly highlight key limitations of the study, including its retrospective design, limited sample size in certain thrombophilia subgroups, absence of routine anti-Xa monitoring, and the single-center setting that may limit generalizability. We have also acknowledged the potential for confounding by indication in LMWH allocation. The revised conclusion addresses existing knowledge gaps, such as the lack of standardized therapeutic algorithms and reliable risk stratification tools. Finally, we emphasize the need for future prospective, multicenter studies with well-defined inclusion criteria, uniform LMWH protocols, and comprehensive maternal and neonatal outcome assessments to better delineate the role of anticoagulation in thrombophilia-affected pregnancies.
Comment 15: The study title should include both prevalence and therapeutic efficacy.”
Response 15: We thank the reviewer for this constructive suggestion. In response, we have revised the title of the manuscript to better reflect the dual focus of the study—namely, the prevalence of pregnancy complications in women with inherited thrombophilia and the evaluation of the therapeutic efficacy of low molecular weight heparin (LMWH). The updated title now reads: “Impact of Inherited Thrombophilia on Pregnancy Complications and the Role of Low Molecular Weight Heparin Therapy: A Case-Control Study” This revised title more accurately conveys the scope and objectives of the study, as recommended.